# Galectin-3 and Epithelial MUC1 Mucin—Interactions Supporting Cancer Development

**DOI:** 10.3390/cancers15102680

**Published:** 2023-05-09

**Authors:** Iwona Radziejewska

**Affiliations:** Department of Medical Chemistry, Medical University of Białystok, ul. Mickiewicza 2a, 15-222 Białystok, Poland; iwona@umb.edu.pl; Tel.: +48-(85)-7485675

**Keywords:** cancer, Gal-3, galectins, glycosylation, MUC1, mucins, TACAs

## Abstract

**Simple Summary:**

MUC1 mucin with T antigen and galectin-3 with high affinity to T disaccharide are both overexpressed in a variety of human cancers. Their mutual interactions support cancer development, attenuate anoikis, promote cancer cells proliferation, invasiveness and metastasis. The inhibition of MUC1/T antigen—galectin-3 interactions may be a potential strategy to reduce tumor progression and metastasis. The aim of this review is to summarize the current knowledge about the relationship between MUC1 and galectin-3.

**Abstract:**

Aberrant glycosylation of cell surface proteins is a very common feature of many cancers. One of the glycoproteins, which undergoes specific alterations in the glycosylation of tumor cells is epithelial MUC1 mucin, which is highly overexpressed in the malignant state. Such changes lead to the appearance of tumor associated carbohydrate antigens (TACAs) on MUC1, which are rarely seen in healthy cells. One of these structures is the Thomsen-Friedenreich disaccharide Galβ1-3GalNAc (T or TF antigen), which is typical for about 90% of cancers. It was revealed that increased expression of the T antigen has a big impact on promoting cancer progression and metastasis, among others, due to the interaction of this antigen with the β-galactose binding protein galectin-3 (Gal-3). In this review, we summarize current information about the interactions between the T antigen on MUC1 mucin and Gal-3, and their impact on cancer progression and metastasis.

## 1. Galectins

Galectins are groups of small, soluble glycan-binding proteins with similar binding affinities for β-galactosides. They share a highly conserved 130 amino acid long, β-sandwich folded sequence in the evolutionarily-conserved carbohydrate recognition domain (CRD) which is responsible for glycan binding [1,2,3,4]. So far, 15 members of mammal galectin family (11 in humans) have been identified [5,6]. They are divided into three different groups: a. prototype single-CRD galectins that are able to form non-covalent homodimers by self-association (Gal-1, -2, -5, -7, -10, -11, -13, -14, -15); b. tandem-repeat galectins with two CRDs connected by an amino acid sequence-embedded linker region (Gal-4, -6, -8, -9, -12); c. chimera-type galectins with a single CRD motif connected to collagen-like N-terminal domain (Gal-3) [6,7,8]. Gal-5, -11, -15 and -6 are not found in humans [9].

Galectins are synthesized in cytoplasm and reside in the cytosol, nucleus or other cellular compartments for much of their lifetime [10]. They interact with cell surface glycans following their non-classical secretion by exocytic pathway [6,11]. It was reported that all cells express galectins, but their expression pattern varies between cell types and tissues [10]. Generally, galectins function by binding to the carbohydrates of glycoproteins and glycolipids on the cell surface. It has been also reported that intracellular galectins can bound to glycans displayed on damaged intracellular organelles [8]. The ligands of galectins are glycoproteins and glycolipids with different degrees of oligosaccharide modifications [12]. Lactose (Gal-β(1-4)-GlcNAc) seems to be the minimal carbohydrate ligand necessary for binding to galectins. Most structures of galectins have been reported with lactose bound. It was noted that the 3-OH group is crucial for sugar recognition, and the substitution of 4-OH and 6-OH groups on the galactose ring mostly attenuates binding. These mentioned hydroxyl groups in lactose form hydrogen bonds with side chains of hydrophilic residues from galectins. Disaccharide is effectively “captured” by the galectin peptide loop above the lactose molecule and flat side chain at the bottom of the disaccharide [13]. 

The ubiquity of galectins in both normal and cancerous cells suggests their crucial role in cell functions, likely by interactions with omnipresent galactose-containing glycoforms. Galectins mediate the functions of the cells both intracellularly and extracellularly. Generally, their actions include the regulation of cell growth, pre-mRNA splicing, cell-cell and cell-extracellular matrix adhesion, cellular polarity, motility, cell migration, differentiation, transformation, signal transduction and apoptosis [9,14,15]. The total function of any galectin can vary noticeably and their activities can be multi-faceted. Galectins self-association and interplays with cell surface glycoforms, as well as interactions with other biomolecules, both intracellularly and extracellularly, can have an important impact on galectin functions. In relation to cancer, numerous investigations have uncovered the various roles and mechanisms of actions of galectins in tumor cell invasiveness and dissemination. Dysregulation of their expression and correlation with aggressiveness has been frequently observed in many types of cancers [13,15]. Thus, the diversity of galectin functions becomes an attractive subject for studying in medicine and pharmacy.

### Galectin-3

Galectin-3 (Gal-3) (Mac-2, IgE-binding protein, L-29, LGALS3 or CBP30) is 29–35 kDa β-galactoside-binding protein belonging to chimera-type galectins with a single CRD motif (130 amino acids) and a non-lectin N-terminal domain (NTD) (110–130 amino acids) with a collagen-like linker region rich in glycine, tyrosine, and proline residues (Figure 1) [4,16,17,18]. 

The N-terminal domain is responsible for galectin-3 multimerization upon contact with multivalent ligands and has the unique ability to form pentamers or other oligomers which allow the creation of lattices with glycoproteins and glycolipids [19,20]. It can act as a bivalent or multivalent ligand [20,21]. Multimerization is a common feature of extracellular lectin, where in this form its triggers the initiation of cell surface molecule-associated cell signaling [20,22,23,24,25]. Multimerization of galectin-3 often points to its activity in cancer and inflammation processes [26]. Gal-3 was initially described to bind especially to type 1 or 2 Galβ1-3(4)GlcNAc (N-acetyllactosamine) chains, and its affinity increases for polylactosamine structures and branched glycans over simple carbohydrates. For example, lactose and N-acetyllactosamine are much stronger ligands for galectin-3 than galactose, and also N-acetyllactosamine has higher affinity for galectin than lactose [27,28,29]. Such specific bounding may be enhanced or attenuated depending on the substituents that modify subterminal galactose residues in the galectin ligand [30]. It has been also reported that Gal-3 can interact with many intracellular proteins by protein-protein interactions in a carbohydrate-independent manner [27,31,32]. Galectin-3 is commonly expressed in immune cells, epithelial cells, and endothelial cells. Depending on the cell type, Gal-3 can be found mainly in the cytoplasm, nuclei, and mitochondria. Additionally, it is secreted to the cell surface and into biological fluids as a soluble protein through a non-classical exocytosis process [1,6,7,8,16,21,27]. Galectin-3 undergoes posttranslational modifications, such as limited proteolysis and phosphorylation. Extracellular Gal-3 may be cleaved at the alanine 62/tyrosine 63 position by various molecules, such as metalloproteases (MMPs) and the prostate-specific antigen (PSA) [30]. Tyrosine phosphorylation of galectin-3, mediated by c-Abl kinase, seems to be crucial for its motility, lysosomal degradation, cleavage inhibition, and secretion. Serine phosphorylation can be associated with diverse binding of Gal-3 to laminin and mucin [33,34,35]. The distribution of the lectin can be changed with cellular proliferation, differentiation, and development [30]. 

Gal-3 functions result from protein-protein interactions, showing its versatility [30]. A number of important roles in cancer initiation and progression, as well as in tumor-immune escape, have been associated with this lectin [22,36,37]. Intracellularly, galectin-3 is involved in numerous cellular functions as it acts as a multifunctional oncogenic protein that can associate with Ras or Bcl-2 to help regulate cell growth and apoptosis [13,38,39]. Gal-3 was reported to have similar antiapoptotic action to that of Bcl-2 [40]. Galectin regulates Bcl-2 and other Bcl-2 family members by direct binding of these molecules as Gal-3 contains the NWGR motif found in the BH1 domain of the Bcl-2 protein [18,41]. Gal-3 is associated with Ras signaling as it interacts by its CRD with activated K-Ras. The consequence of such interplay is enhanced translocation of Gal-3 to the plasma membrane and an increase of K-Ras signaling, promoting PI3-K activation. Through the aforesaid relationship, Gal-3 and Ras participate in the regulation of such processes as proliferation and survival of cancer cells [30,31]. Apart from that it interacts with annexin VII, a Ca^+2^ and phospholipid-binding protein, mediating Gal-3 translocation to the perinuclear mitochondrial membrane, where it controls mitochondrial integrity and cytochrome c release, which is essential for apoptosis regulation [39,42]. In the nucleus, galectin-3 promotes pre-mRNA splicing and participates in spliceosome assembly via complexes with nuclear protein Gemin4 [4,23]. It can also regulate gene transcription by augmenting transcription factor association with Spi1 and CRE parts in the gene promoter sequence [4,43]. Extracellularly, this lectin binds to a large array of glycoproteins and glycolipids on the cell surface and in the extracellular matrix, which play biological roles in cell aggregation, angiogenesis, cell adhesion, immune system evasion, and tumor metastasis. These can be various ligands, such as fibronectin, laminin, vitronectin, elastin, lysosomal-associated membrane protein (LAMP) 1 and 2, neutral cell adhesion molecule (N-CAM), integrin α3β1, CD43, CD45 on leukocytes, CD66, immunoglobulin IgE, IgE receptor, and epithelial MUC1 mucin [6,9,13,21,28]. Enhanced adhesion of Gal-3 to ECM supports the escape of cancer cells from primary tumor sites [27]. Additionally, by cross-linking cell surface glycoconjugates, it can trigger a cascade of transmembrane signaling events [21,44,45]. It was suggested that galectin-3 could promote tumor metastasis mostly in an Akt-dependent way. It has been reported that cooperation between sialyl Tn antigen and galectin-3 has resulted in Akt pathway activation and an increase in the transcription activity of β-catenin and protein synthesis [46,47]. Gal-3 has been also reported to be implicated in epithelial-mesenchymal transition (EMT) in colon cancer patients, where its overexpression negatively correlated with tumor recurrence and survival [48]. Moreover, galectin-3 is engaged in the regulation of main cytokines involved in inflammation, inducing those that have been implicated in cancer, such as interleukin-6 [18,49,50]. Moreover, Gal-3 has been involved in angiogenesis, a crucial step in tumor cell invasion and metastasis. Such effect implies the interaction of galectin with integrins or glycans expressed on cell surface membranes, which results in the clustering of Gal-3 with ligands and the activation of focal adhesion kinase leading to the regulation of VEGF—basic factor-mediated angiogenesis [27,51]. Up to 30-fold enhanced concentration of Gal-3 in blood circulation in patients with different cancers has been reported [37,44,52,53,54]. Additionally, higher concentrations of this galectin are observed in the sera of patients with metastatic disease than in the sera of patients with localized tumors [37,55]. Upregulation of this lectin was revealed in transformed and metastatic cell lines as well as in many human carcinomas, such as breast, colon, gastric, hepatocellular, tongue, leukemia, and in well-differentiated thyroid carcinomas [6,52]. Such increased expression correlated with immune suppression, progressive tumor stages, and metastasis [45,49]. It has been also reported that galectin-3 binds to the discrete sets of glycoproteins on the surface of T cells, and triggers T cell death [56,57,58]. Decreased expression of Gal-3, compared to corresponding normal tissue, has been also reported, e.g., in breast, ovarian, prostate tumors, epithelial skin cancer, and malignant salivary gland neoplasms. It seems that downregulation of Gal-3 expression is specific, especially in the initial stages of cancer development [6]. Due to the significant role of Gal-3 in tumor development, its downregulation and targeting of Gal-3 ligands seems to be a promising approach for cancer therapy [59,60]. 

## 2. Mucins

Mucins (MUCs) are highly O-glycosylated proteins with high molecular weight and complex molecular organization. They are synthesized principally by epithelial cells and provide protection and lubrication to the epithelial surfaces [61,62]. Mucins are generally classified into two subfamilies: secreted and membrane-bound. The first group comprises MUC2, MUC5AC, MUC5B, MUC6, MUC7, MUC8, and MUC19. Membrane-bound mucins include MUC1, MUC3A, MUC3B, MUC4, MUC11-13, MUC15-17, MUC20, MUC21, and MUC22 [61].

Mucins bear a heterogeneous variety of O-glycan structures, which are said to be the most information dense biological macromolecules in animal cells [63,64,65]. Glycosylation is a common post-translational process occurring in ER and the Golgi apparatus. There are about 2000 glycosyltransferases which are involved in this process [66]. Apart from dominating O-GalNAc glycans linked to Ser/Thr of the mucin polypeptide chain (mucin-type O-glycosylation), there are also N-glycans that are linked to Asn residues [64,67]. 

The main function of secretory mucins is the protection of the epithelium, while the roles of membrane-bound mucins include participation in cell adhesion, interaction with a variety of glycan-binding proteins and other cell surface receptors, the regulation of the immune system, as well as cell signaling, growth and proliferation. Due to such functions, epithelial mucins seem to be of particular concern for the biology of cancer cells as they are mostly overexpressed and abnormally glycosylated in a variety of different tumors [61,64]. In cancer cells, the O-glycans are usually truncated and highly sialylated, while N-glycans can be changed in their branching pattern and expose specific epitopes as the Lewis antigens. The mechanism of such changes involves upregulation and downregulation of the expression of proper glycosyltransferases [64]. The aberrant mucin-type O-glycan synthesis pathway can affect the aggressiveness of tumor cells comprising the ability to spread through the circulation and metastasize [68]. 

### MUC1

MUC1 (known also as episialin, EMA, PEM, PUM, CA15-3, KL-6, H23Ag, and MAM6), one of the best studied type I transmembrane mucins, is physiologically expressed on the apical surface of the most secretory epithelia, including those in the mammary gland, and the gastrointestinal, respiratory, urinary, and reproductive tracts [37,69]. It is synthetized as a single polypeptide and then autocleaved in the endoplasmic reticulum into longer N- and shorter C-terminal subunits. These subunits form a stable complex held together through a noncovalent interaction on the cell surface (Figure 2) [21,69,70,71,72,73]. 

Extracellular domain can be released by inflammatory stimuli such as interferon gamma (INF-γ) and tumor necrosis factor alpha (TNF-α) upon the action of specific enzymes ADAM17 (disintegrin and metalloprotease domain containing protein-17) or matrix metalloproteases (MMPs); and can be found in biological fluids such as serum, the lumen of the intestinal tract, and culture supernatants of mucin expressing cells [69,70,74]. Excessive shedding of this domain is often observed for metastatic carcinoma [75]. The N-terminal (MUC1-N) extracellular domain of MUC1 extending up to 200–500 nm out of the cell surface contains highly polymorphic sequence motifs, the variable number of 20-amino acid tandem repeat regions (VNTR) rich in serine, threonine and proline (PDTRPAPGSTAPPAHGVTSA), and the SEA (sea urchin sperm protein enterokinase and agrin) domain [71,76]. Five serine/threonine residues of VNTR region are possible O-glycosylation sites (mucin-type O-linked glycosylation). Proline residues give rigidity and contribute to a highly extended protein structure [77]. N-glycosylation sites are usually present outside of this region. The SEA domain has the proteolytic cleavage site, and some amino acid sequences of this region are important for the non-covalent association of protein subunits. The extended structure of the extracellular domain seems to act as a kind of particularly efficacious frame for the presentation of oligosaccharide chains to lectins, including Gal-3 [54]. Moreover, it functions as a cell barrier, blocking cell-cell and cell-extracellular matrix interactions and protecting cells from cellular and pathogenic invasions while keeping the epithelium intact [71,78]. C-terminal (MUC1-C) domain consists of a short 58 amino acid extracellular domain (ECD), a 28 amino acid transmembrane domain (TM), and a 72 amino acid cytoplasmic tail (MUC1-CT) [39,56,64]. This highly conserved cytoplasmic domain contains seven tyrosine residues and several serine and threonine residues, which represent putative recognition sites for receptor tyrosine and other kinases, such as epidermal growth factor receptor (EGFR), glycogen synthase kinase 3β (GSK3β), or protein kinase C delta (PKCδ). The function of this cytoplasmic domain is more related to signal transduction [79,80]. The phosphorylated MUC1-C domain binds directly to the PI3K SH2 domain and activates the AKT→mTOR pathway. It also plays a role in stimulating MEK/ERK signaling and the suppression of the RASSF1A tumor suppressor, which impedes the RAF/MEK/ERK pathway and additionally is one of the most frequently inactivated genes in human cancers [74,81,82]. Additionally, MUC1-CT is also demonstrated to regulate the activity of the nuclear factor kappa B (NF-κB) pathway in breast cancer by cooperating with and stimulating IkB kinase (IKK) family members and NF-κB p65 [77,83,84,85].

The epithelial MUC1 mucin has been found to be greatly overexpressed and abnormally glycosylated in most tumor cells studied, and its expression level is correlated with a poor prognosis [21,86]. Such changes provide tumor cells with invasiveness, metastasis, and resistance to death, generally with complex mechanisms [71,87]. Malignant MUC1 is involved in multiple signaling pathways which are associated with various aspects of tumor progression [69,75]. It has been shown to interact via its cytoplasmic domain with essential intracellular proteins, including β-catenin and p53, which is involved in signal transduction and apoptosis regulation in response to DNA damage [37,88]. Overexpression of MUC1 is due to the loss of polarity in epithelial cells. It becomes expressed over the entire surface [37]. Apart from that, the glycosylation pattern of the extracellular domain of MUC1 in cancers differs distinctly from that of MUC1 expressed on healthy cells. The long-branched carbohydrate side chains become uncompleted and new, unique sugar antigens called tumor associated carbohydrate antigens (TACAs) are formed [69,71,89,90,91]. It is said that such antigens are ideal targets for anti-tumor immune prevention and therapy due to the lack of their expression in healthy cells [56,92]. The most common TACAs are especially GalNAcα Ser/Thr (Tn antigen), sialylα2-6GalNAc (sialyl Tn), Galβ1-3GalNAcα Ser/Thr (T antigen, oncofetal Thomsen-Friedenreich (TF) antigen), and sialylα2-3Galβ1-3GalNAcα Ser/Thr (sialyl T). Additionally, more complex, neutral, sialylated and fucosylated glycans, and those carrying Lewis epitopes like sialylα2,3-Galβ1,-3-(Fucα1,-4)-GlcNAc-R (sialyl-Lewis^a^), and sialylα2,-3-Galβ 1,-4(Fucα1,-3)-GlcNAc-R (sialyl-Lewis^x^) are found in cancer cells [64,93,94,95]. High sialylation can cause premature termination of chain elongation, and the formation of above-mentioned truncated sugar antigens [69,71]. The mechanisms of such highly specific alterations in the glycosylation pattern include alterations in the mucin core peptide, mucin subcellular localization or expression, and the upregulation and downregulation of the expression of glycosyltransferases involved in the synthesis of glycans [61,64,96,97]. Many authors have reported a positive correlation between the occurrence of tumor-associated antigens and tumor progression, and as a consequence, a poor prognosis and reduced overall survival in various types of cancer [19,89,95,98,99]. Such changes in glycosylation impact the stability and subcellular localization of MUC1. Compared with physiological, fully glycosylated mucin, hypoglycosylated MUC1 may enhance MUC1 oncogenic signaling by decreasing its cell surface levels and increasing intracellular accumulation [69]. Apart from that, incomplete O-glycan side chains leading to the production of underglycos1ylated forms of MUC1 reveal immunogenic epitopes which can generate the immune response and cancer-related inflammation [100]. It is also suggested that hypoglycosylation unmasks the peptide core of mucin, allowing extracellular domain cleavage and release by the action of proteases [63,69,101]. As mentioned earlier, MUC1 is able to regulate p53 responsive gens, and, by that, cell fate [88,102]. MUC1 directly binds to the tumor suppressor p53 regulatory domain, and selectively stimulates the transcription of growth arrest genes and decreases the transcription of apoptotic genes, and thereby decreases cell death. Additionally, MUC1 stimulates the anti-apoptotic Bcl-xL and PI3/Akt pathway to attenuate apoptosis [102,103]. It has also been demonstrated that direct binding of MUC1 with caspase-8 and the death effector domain of FADD prevented the activation of the death receptor-induced extrinsic apoptotic pathway [104]. MUC1, over-expressed in the cancer state, stimulates tumor cell release from initial tumor sites by suppressing E-cadherin-mediated cell-cell and integrin-mediated cancer-extracellular matrix interactions, promoting metastasis [69,105,106]. Recently, it has been demonstrated that tandem repeats regions of MUC1 are able to activate NF-κB, a transcription factor implemented in pro-inflammatory responses, the induction of resistance to chemotherapy, tumor progression, invasion, and metastasis [83,107]. Moreover, it is hypothesized that altered glycosylation enables cancer MUC1 to function as a ligand for cell adhesion molecules such as selectins, epidermal growth factor receptor (EGFR), intercellular adhesion molecule-1 (ICAMs), and ECM components, aiding adherence of MUC1-expressing circulating tumor cells to endothelial cells and seeding at isolated sites that settle secondary tumors. An increase in MUC1—EGFR interaction leads to the activation of this factor, which is likely associated with tumorigenesis and cancer progression [63,106,108,109]. Thus, the results of such interactions are inherently associated with the induction of invasion, migration, metastasis, angiogenesis, and the inhibition of apoptosis [69,105,110,111,112]. Based on cancer cell specificity and its special roles in carcinogenesis, MUC1 is considered one of the most promising targets and biomarkers in cancer research [65,113]. 

## 3. Gal-3 and MUC1 Relationships

The studies show that cancer-associated MUC1 mucin is one of the most desirable, natural ligands for galectin-3, and that such interactions are mediated mostly via the binding of Gal-3 to the T carbohydrate (Figure 3), a prominent tumor-associated antigen on MUC1 mucin. 

The sugar part of MUC1 seems to be easily accessible to galectin-3 due to its rod-like, rigid structure, which is longer (200–500 nm) than common cell surface adhesion molecules (~30 nm) [21]. It has also been demonstrated that the binding affinity of Gal-3 for oligosaccharides of MUC1 increases greatly by the clustering of its protein [114]. The cross-linking of multivalent glycoconjugates and specific receptors can lead to increases in galectin-binding affinity [19]. Kinetic analysis revealed that the binding affinity of galectin-3 to the T antigen on MUC1 showed to be more than 5-times higher than to free T disaccharide [19,109]. Although the binding of galectin-3 to other cell-surface ligands is of course possible, MUC1 is considered as the most significant target known [28,106]. There are reports about the importance of the hydrogen bonds network in interactions between Gal-3 and T disaccharide [19,115]. T antigen is linked with the protein backbone of MUC1 via serine or threonine residues, and it was reported that galectin interacts with this protein backbone, which also augments the binding. Transitory interaction between amino acid threonine and galectin-3 has been revealed [109,116,117]. It has been demonstrated that Gal-3 coincided with that of MUC1 on the surface of various human tumor tissues, but not in human nonmalignant cells. Moreover, the level of galectin maintained on the surface of different cancer cells paralleled that of mucin [21]. It is believed that interactions of Gal-3 with MUC1 via T antigen influence a number of key steps in cancer progression and metastasis [19,27,37,56]. T antigen (Galβ1-3GalNAcα-Ser/Thr; CD176; the Thomsen-Friedenreich (TF) antigen), which is covered by more expanded glycosylation and sialylation in normal epithelium, found in ~90% of all human cancers, is an intermediate structure in the biosynthesis of O-complex O-linked oligosaccharides on MUC1 [118]. Its creation is carried out by the addition of galactose from UDP-Gal to the precursor, Tn antigen (GalNAcα-Ser/Thr) by core 1 β1,3-galactosyltransferase (C1GALT1, T synthase) [19,99,115,119]. It is suggested that the unbalanced expression of such specific glycosyltransferases implemented in the glycosylation pathways, as well as altered availability of precursor monosaccharide molecules, are responsible for altered glycosylation patterns in cancers, and thus are likely the key factors responsible for the enhanced viability of T disaccharide [19,118,120]. Apart from that, it was reported that the availability and activity of the molecular chaperone Cosmc (Core 1 β3-Gal-T-specific molecular chaperone), which is responsible for the proper folding of a functional C1GALT1, and prevents its ubiquitin-mediated proteasomal degradation, can also influence the total T antigen appearance in many cancers [19,63,119,121,122,123,124]. Additionally, it is worth emphasizing that in normal epithelium, T antigen is covered by extensive glycosylation, sialylation or sulphation, but expressed in an uncovered form by most human cancer cells [108,118,125]. 

It has been demonstrated that interaction between MUC1 with unsubstituted T antigen and galectin-3 resulted in MUC1 cell surface clustering and the exposition of smaller cell surface adhesion molecules, such as E-cadherin. Such aggregation enhanced the survival of the cells by preventing cellular anoikis, a specific type of apoptotic process which is induced by the loss of cell adhesion or a deficient cell-matrix interactions, and is proposed to be the leading mechanism removing disseminating cancer cells from the circulation. Moreover, aggregated cells have been observed to have a much higher survival rate in the circulation than single cells. Thus, increased survival of the spread tumor cells caused by galectin-3-MUC1 mucin interactions may have serious consequences on the metastatic potential of the cancer cells as it prolongs the survival of disseminated tumor cells in the circulation. Apart from that, enhanced cancer cells aggregation, as the result of mucin-galectin interactions, is likely to increase the physical trapping of circulating cancer cells in the microvasculature at target tissues, which also intensifies metastasis [108,126]. The increased expression of Gal-3 in the bloodstream of many cancer patients stimulates several important steps of the metastatic cascade. There are also reports demonstrating that the binding of galectin-3 to the T antigen on MUC1 mucin activated the MAPK and PI3K/Akt signaling pathway, which consequently led to the intensification of cell proliferation and motility [21].

The large size and length of MUC1 enables it to form a kind of protective shield on the cell surface, and in such a way inhibit cell-cell and cell-extracellular matrix interactions [106]. Thus, the binding of galectin-3 to the T antigen on the MUC1 extracellular domain promotes MUC1 cell surface polarization, which results in the exposure of adhesion molecules that were previously covered by the large and heavily glycosylated cell surface MUC1. The “Protective shield” of MUC1 may be broken, smaller cell surface adhesion ligands can be revealed, and cancer-endothelial adhesion can be enhanced, which can lead to promoting metastasis. The survival of disseminating tumor cells in the circulation is possible by preventing the initiation of anoikis [27]. Thus, the protective effect of the MUC1 barrier and the de-protective outcome of the galectin-3-T antigen/MUC1 interaction on cancer cell adhesion give explanations at the molecular level for many late clinical and experimental studies related to metastasis; e.g., the correlation between increased apical MUC1 cell surface polarization and increased lymphatic invasion, recurrence rate, and lower overall survival in patients with breast cancers [106,127]. A similar pattern of interactions has been observed for Gal-3 and MUC4 mucin. The binding of galectin-3 resulted in mucin clustering that exposed different adhesion molecules (e.g., integrins) that were masked by MUC4. This could facilitate the attachment of tumor cells to endothelial cells, a crucial step for the discharging of circulating tumor cells [63,128]. It has also been demonstrated that improved prognosis in gastric cancer was correlated with an increased concentration of circulating anti T-antibodies, which would inhibit galectin-3-mediated T antigen/MUC1 interactions [129]. Moreover, there have been interesting reports about the association of MUC1 sialylation with a better prognosis in breast cancer as a result of inhibiting MUC1-galectin-3 interaction. Such inhibition resulted from hiding T antigen by sialic acid, which reduced galectin-3-T antigen/MUC1 interactions [37]. Suggested models of Gal-3—MUC1 interactions highlight the functional importance of the change of the cell surface glycosylation in cancer progression and metastasis.

There are also demonstrations indicating that MUC1-C associates with Gal-3 intracellularly. Glycosylation of MUC1-C on Asn-36 functions as a binding site for galectin-3. Both MUC1-C and galectin-3 bind directly to β-catenin. Apart from that, Gal-3, such as MUC1-C, localizes to the nucleus and coactivates β-catenin/Tcf4-mediated gene transcription [130]. However, there are also reports which claim that there is no binding of Gal-3 with the MUC1 C-terminal subunit. Tanida et al. [54] noted that the binding of Gal-3 to the MUC1 extracellular domain triggered the recruitment of β-catenin to MUC1-C. Summarized consequences of antigen T/MUC1—galectin-3 interactions are presented in Figure 4.

It has been suggested that MUC1 and galectin-3 expression could be coordinately regulated [39]. It was revealed that cancer cells with gain and silencing of MUC1 indicated that MUC1 upregulates Gal-3 expression at the mRNA level. The authors demonstrated that MUC1-C is able to induce Gal-3 expression by a posttranscriptional mechanism, and also that N-glycosylation of MUC1-C Asn-36 seems to be essential for such a response. The deglycosylation of MUC1-C canceled the binding of mucin with Gal-3, indicating that their relationship is not mediated by protein-protein interplay. Other findings proposed that galectin acts in crosslinking MUC1 to EGFR and likely other cell surface receptors. Li et al. reported that MUC1 and EGFR associated inherently at the cell membrane, and they suggested that Gal-3 might function as a bridge to physically associate MUC1 with EGFR [132]. Thus, the blocking of Gal-3-binding with lactose, or the silencing of Gal-3 with siRNA, were linked with the suppression of MUC1-EGFR complex formation in the response to EGFR stimulation. Apart from that, the stimulation of EGFR also induces the binding of the MUC1-C to β-catenin, and such a response is related to the Gal-3-mediated interaction between MUC1 and EGFR [39,54,132,133].

## 4. MUC1- Gal-3 Inhibitors as a Therapeutic Strategy

The appearance of the cancer-associated carbohydrate T antigen on MUC1 mucin overexpressed in cancer cells, and the increased expression of galectin-3 are both typical features in various cancers. Thus, the influence of MUC1/T antigen—galectin-3 interactions on cancer development raises promising therapeutic strategies involving the inhibition of such interactions [19]. Several possibilities have been proposed. These include, e.g., using anti-TF antibodies or TF-mimicking peptides leading potentially to blocking a primary metastatic step and providing a survival advantage [134]. Moreover, applying negative galectin-3 mutants—NH(2)-terminally truncated form of Gal-3 (galectin-3C)—resulted in reducing metastases, tumor volumes and weights in primary tumors in an orthotopic nude mouse model of human breast cancer [135]. Apart from that, synthetic or semi-synthetic oligosaccharides have been identified as a promising new class of therapeutic agents to target Gal-3-mediated metastasis [136,137].

## 5. Conclusions

The relationships between MUC1 mucin with T antigen and galectin-3 seem to be very important in cancer development. Thus, understanding the issue give opportunity to develop new therapeutic strategies based on inhibiting of such interactions. I believe that presented review well summarize the current knowledge in the subject.

## Figures and Tables

**Figure 1 cancers-15-02680-f001:**
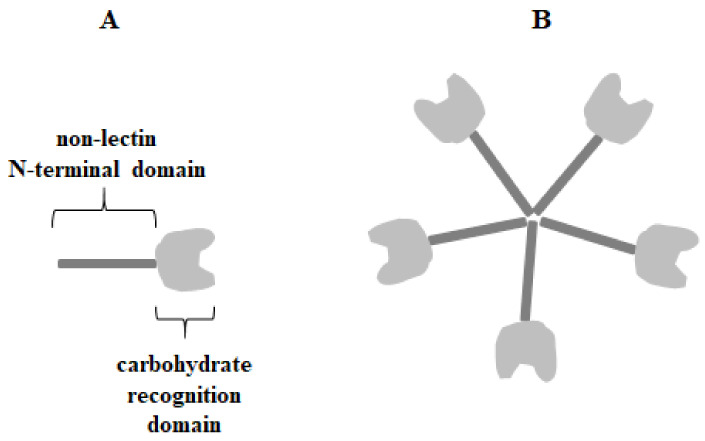
T The structure of chimera-type galectin-3 in form of monomer (**A**) and pentamer (**B**).

**Figure 2 cancers-15-02680-f002:**
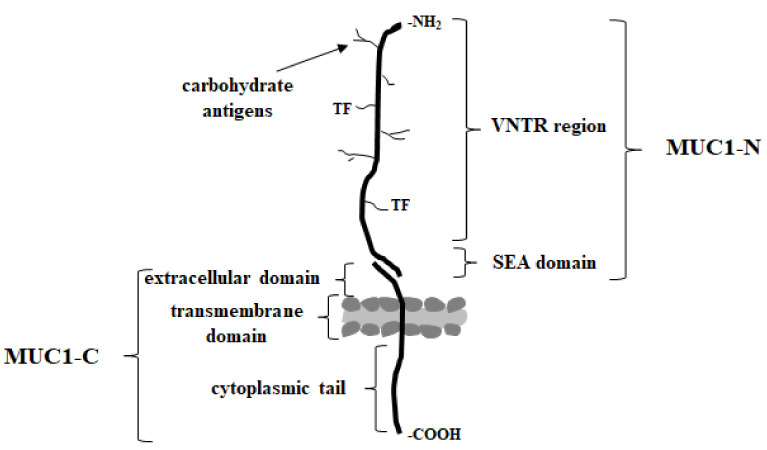
The structure of underglycosylated, cancer MUC1 mucin.

**Figure 3 cancers-15-02680-f003:**
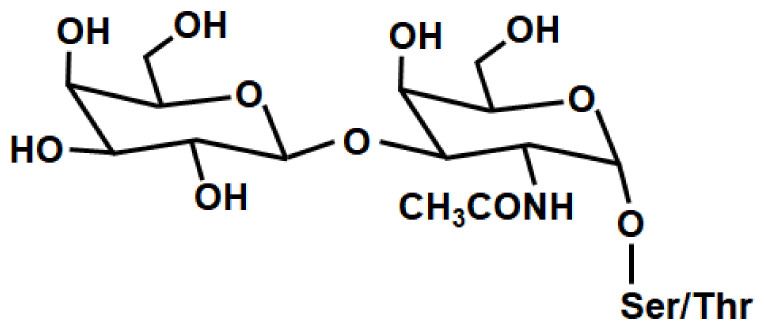
The structure of T antigen.

**Figure 4 cancers-15-02680-f004:**
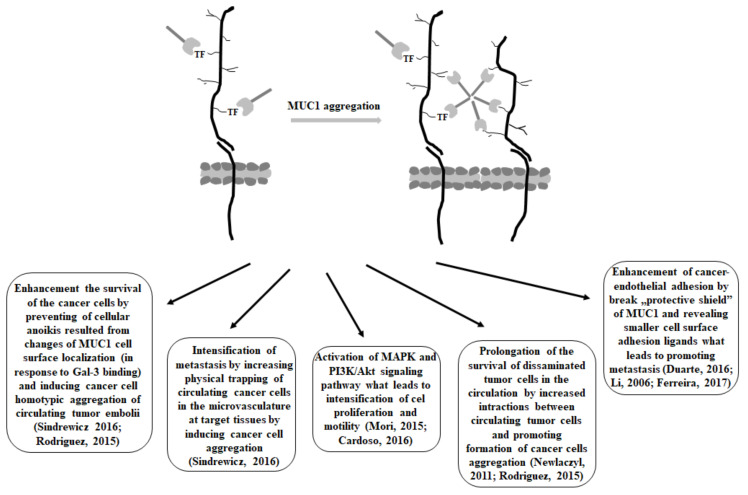
T Galectin-3 interactions with TF antigen on MUC1 and possible consequences of such interactions [19,21,27,30,100,109,127,131].

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
