# Peer review of "Galectin-3 and Epithelial MUC1 Mucin—Interactions Supporting Cancer Development"

_cancers, 2023, doi:10.3390/cancers15102680_

Round 1

Reviewer 1 Report

The submission of Radziejewska is a review focused on the relation and importance of Galectin-3 and Mucin-1 (MUC1) interactions.

MUC1 is a glycoprotein extensively studied, mainly because MUC1 in tumor cells present a truncated glycans, such as TF-antigen. TF-antigen is an important ligand for Galectin-3 and this interaction is associated with tumor progression and metastasis. This review described and summarized the information about the relationship of Galectin-3/MUC1 and their implications on the health and cell life cycle.

The organization is logical. This work will be of interest to the scientific community, especially given the role of this type of recognition in new anti-cancer therapies.

Minor revisions:

Page 3 line 90 – “Glectin-3” replace by Galectin-3

Page 3 line 129 – the reference 48050?

Page 4 lines 166-168 – should be eliminated.

Page 6 line 227-229 – replace “Sr” by Ser in all antigens.

Page 6 line 231-  replace “sialylα2,3-Galβ1,3-(Fucα1,3)-GlcNAc-R (sialyl-Lewisx)” by sialylα2,3-Galβ1,4-(Fucα1,3)-GlcNAc-R (sialyl-Lewisx)

Page 7 line 288 - replace “Sr” by Ser

Should be included a figure with the structure od the TF-antigen.  

Author Response

Thank you for valuable remarks considering the manuscript.

All the corrections were introduced to the text. The added parts are seen in red. The structure of TF antigen was added to the text (Figure 3).

Reviewer 2 Report

In the review entitled ‘Galectin-3 and epithelial MUC1 mucin – interactions supporting cancer development ‘ the author provides a clear synthesis of reported interactions between galectin-3 and MUC1 in literature and the involvement of these interactions in carcinogenesis. The manuscript is well organized : the author describes in a first part galectins and in a second one mucins and finishes on the interactions between galectin-3 and MUC1.

-Concerning the form of the manuscript:

-               * About the section labelling: there is 1.1. galectin-3 but no 1.2 section

-              * In the same way, there is 2.1 MUC1 but no 2.2 section

-         * There is a sentence, lines 166-168, which should be removed. I think it corresponds to author instructions

- In galectin-3 section, the author should give more details about the mechanism involved in galectin-3’s function in carcinogenesis (brief description of mechanisms of galectin-3 involvement in cancer through cell cycle regulation, resistance to apoptosis, angiogenesis…)

-In mucins section, the author should add other mucin modifications involved in carcinogenesis such as quantitative and qualitative alterations (alternative splicing, delocalization, glycosylation…). A brief synthesis of the mechanism of mucins involvement in cancer (before 2.1) should be useful (immunity modulation, interaction with the environment, apoptosis, cell signaling, chemoresistance...)

 - in figure 3, The author should add details in the boxes about the mechanism contributing to the effect: for example, how galectin-3-MUC1 interaction contribute to « prolongation of the survival of disseminated tumor cells in the circulation ». A reference should also be mentioned in each box.

-line 319-330: for this mechanism, it could be mentioned that MUC4 and galectin-3 interaction results in the same consequence, that is to say clustering of MUC4 and exposition of adhesion molecules.

 -The author should provide a section about inhibitors reported in literature to prevent the interaction between MUC1 and Galectin-3 and their potential uses in cancer treatment.

Author Response

Thank you for valuable remarks considering the manuscript.

Specific answers:

  * About the section labelling: there is 1.1. galectin-3 but no 1.2 section

 * In the same way, there is 2.1 MUC1 but no 2.2 section

Answer: Corrected

 * There is a sentence, lines 166-168, which should be removed. I think it corresponds to author instructions

Answer:  the sentences in lines 166 – 168 were eliminated

In galectin-3 section, the author should give more details about the mechanism involved in galectin-3’s function in carcinogenesis (brief description of mechanisms of galectin-3 involvement in cancer through cell cycle regulation, resistance to apoptosis, angiogenesis…)

Answer:  Some information was added according to suggestions

In mucins section, the author should add other mucin modifications involved in carcinogenesis such as quantitative and qualitative alterations (alternative splicing, delocalization, glycosylation…). A brief synthesis of the mechanism of mucins involvement in cancer (before 2.1) should be useful (immunity modulation, interaction with the environment, apoptosis, cell signaling, chemoresistance...)

Answer: Some information was added according to suggestions

in figure 3, The author should add details in the boxes about the mechanism contributing to the effect: for example, how galectin-3-MUC1 interaction contribute to « prolongation of the survival of disseminated tumor cells in the circulation ». A reference should also be mentioned in each box.

Answer: Added according to suggestion

line 319-330: for this mechanism, it could be mentioned that MUC4 and galectin-3 interaction results in the same consequence, that is to say clustering of MUC4 and exposition of adhesion molecules.

Answer: information about interaction between MUC4 – Gal-3 was added

The author should provide a section about inhibitors reported in literature to prevent the interaction between MUC1 and Galectin-3 and their potential uses in cancer treatment.

Answer: Short section, at the end of an article was added

Reviewer 3 Report

This review comprehensively describes the interaction between Galectin-3 and Epithelial MUC1 mucin from different angles.

One is the modulation of cell adhesion, as galectin-3 can promote cell adhesion by binding to MUC1 mucin. This interaction has been shown to play a role in the metastasis of cancer cells, as the overexpression of MUC1 mucin can increase the ability of cancer cells to adhere to and invade other tissues. Second, galectin-3 and MUC1 mucin interaction can also modulate the immune response. 

Overall, this review paper will broaden readers who are more interested in glycosylation. 

Author Response

Thank you very much for the review of my article